# Evaluation of the Sterile Insect Technique for the Control of the Melon Fly (*Bactrocera cucurbitae*) Under Laboratory and Semi-Field Conditions in Sri Lanka

**DOI:** 10.3390/insects16010021

**Published:** 2024-12-28

**Authors:** Muditha Nawarathne, Lahiru Udayanga, Hasini Ekanayake, Bader Alhafi Alotaib, Ananda Pathirage, Nayana Siriwardena, Athula Jayarathne, Hassan Ammouneh, Mohamed M. M. Najim, Abou Traore, Tharaka Ranathunge

**Affiliations:** 1Hayleys Sunfrost (Pvt) Limited, Biyagama Export Processing Zone, Biyagama 11650, Sri Lanka; mudithagayansw@gmail.com (M.N.); ananda.pathirage@hjs.hayleys.com (A.P.); nayana.siriwardane@hjs.hayleys.com (N.S.); athula.jayaratne@hjs.hayleys.com (A.J.); 2Department of Biosystems Engineering, Faculty of Agriculture and Plantation Management, Wayamba University of Sri Lanka, Makandura 81070, Sri Lanka; 3Department of Agricultural Extension and Rural Society, College of Food and Agriculture Sciences, King Saud University, Riyadh 11451, Saudi Arabia; 4Faculty of Agriculture, Sultan Sharif Ali Islamic University (UNISSA), Kampus Sinaut, Km 33, Jln Tutong, Kampong Sinaut, Tutong TB1741, Brunei; hassan.ammouneh@unissa.edu.bn (H.A.); najim.mujithaba@unissa.edu.bn (M.M.M.N.); 5Department of Community Sustainability, College of Agriculture and Natural Resources, Michigan State University, 328 Natural Resources Building, East Lansing, MI 48824, USA; 6Department of Zoology, Faculty of Science, Eastern University, Chenkalady 30350, Sri Lanka

**Keywords:** *Bactrocera cucurbitae*, control, melon fly, Sterile Insect Technique (SIT), Sri Lanka

## Abstract

The melon fly, *Bactrocera cucurbitae*, threatens many cash crops in Sri Lanka, including gherkin, pumpkins, and melons. This study aimed to determine the best irradiation dose to induce sterility in melon flies, aiming to establish a Sterile Insect Technique (SIT)-based control strategy. Melon fly colonies were maintained in the lab, and male pupae were exposed to different gamma radiation levels (0 to 110 Gy). The effect of radiation on pupal survival, flight ability, reproduction, and lifespan was investigated using standard methods. The 70 Gy dose resulted in the highest pupal survival and flight ability, along with very low fertility and satisfactory lifespan. Furthermore, the competitiveness of the irradiated males (at 70 Gy) against wild males also remained satisfactory under both laboratory and semi-field settings. Therefore, irradiation of *B. cucurbitae* pupae at 70 Gy radiation could be recommended as the best dose to establish SIT programmes in Sri Lanka.

## 1. Introduction

The SIT has emerged as a revolutionary and environmentally sustainable method for controlling insect pest populations, offering a targeted and species-specific approach that minimizes the use of chemical pesticides. Originally proposed by Edward F. Knipling in the 1930s, the SIT has since evolved into a sophisticated and effective tool for integrated pest management [1]. The fundamental principle underlying the SIT involves mass rearing, sterilization, and systematic release of sexually sterile insects into the wild population. When these sterile insects mate with their fertile counterparts, no viable offsprings are produced, resulting in a gradual reduction in the target pest population [1].

The SIT has been successfully applied to control a wide range of insect pests globally, demonstrating its versatility and adaptability across different ecosystems. Significant success stories include the eradication of the screwworm fly (*Cochliomyia hominivorax* (Coquerel 1858)) in North and Central America [2] and the suppression of the Mediterranean fruit fly (*Ceratitis capitata* (Wiedemann 1824)) populations in various regions of the world [3]. Therefore, the SIT has been proven to be effective in cases where traditional control methods, often chemical-based control approaches, face challenges such as resistance development, environmental concerns, and impacts on non-target species.

The SIT has emerged as a promising approach for controlling agricultural pest populations, such as *Bactrocera cucurbitae* (Coquillett 1899). The melon fly was initially reported in 1984 from the Solomon Islands and is currently distributed all over the world. The oriental melon fly, *B. cucurbitae*, poses a severe threat to agricultural productivity in many countries, particularly due to its impacts on the cultivation of cucurbitaceous crops such as pumpkins, cucumbers, and melons [4]. According to previous studies, the melon fly is reported to damage over 81 plant species, while plants belonging to the family Cucurbitaceae are mostly preferred [5]. Bitter gourd (*Momordica charantia*), cucumber (*Cucumis sativus*), muskmelon (*Cu. melo*), pumpkin (*Cucurbita maxima*, *Cuc. pepo* and *Cuc. moschata*), snake gourd (*Trichosanthes anguina* and *T. cucumeria*), and snap melon (*Cu. melo var. momordica*) are some of the most preferred vegetable host plants of the melon fly. Infestation of such a wide range of host plants and the tendency to damage the entire plant (fruits, flowers and stems) have led *B. cucurbitae* to become a major agricultural pest of economic importance that threatens food security [6].

In the Sri Lankan context, *B. cucurbitae* was first reported in the 1950s [4]. The diverse agroecosystems in Sri Lanka have enabled the melon flies to spread geographically and establish themselves, becoming a prominent agricultural pest causing substantial economic losses. The agricultural sector in Sri Lanka plays a crucial role in the country’s economy, with the export of gherkin to various nations being a significant contributor. However, the stringent quality checks imposed during the inspection process pose a considerable challenge, particularly when the presence of melon fly worms inside gherkin containers leads to the rejection of the entire shipment. This not only results in financial losses for Sri Lankan industries, but also underscores the pressing need for the development of an environmentally friendly method to eradicate melon flies from agricultural fields [7].

Despite the use of insecticides, pheromone traps, cultural practices, and biological control approaches, Sri Lanka faces the challenge of mitigating the impacts of *B. cucurbitae* on the agriculture sector [7]. Further, the development of pesticide resistance and environmental concerns associated with chemical interventions underscore the urgency of adopting eco-friendly alternatives for the control of *B. cucurbitae* [2]. Therefore, introduction of the SIT approach represents a strategic and sustainable approach to pest management, with the potential to significantly reduce or even eliminate the population of *B. cucurbitae*. The SIT involves mass production and release of sterile *B. cucurbitae* into the target population, which, when mating with wild counterparts, results in infertile eggs, thereby curtailing the reproductive capacity of the wild *B. cucurbitae* population [8]. The development of the SIT for controlling *B. cucurbitae* in Sri Lanka is grounded in the principles of integrated pest management (IPM) and offers a valuable alternative to conventional chemical-based control methods. However, the successful implementation of the SIT requires a comprehensive understanding of the biology, behaviour, and ecological interactions of *B. cucurbitae* within the local ecosystems.

The establishment of the SIT as a viable strategy to combat the melon fly pest requires the optimization of the minimum dose required to irradiate male melon flies [1]. Assessment of how irradiation impacts the key performance parameters such as survival, flight ability, reproductive capacity, and competitiveness of male *B. cucurbitae* is a key requirement in this process. However, the attention placed on introducing the SIT to control *B. cucurbitae* in Sri Lanka is highly limited, and detailed assessments of the effective irradiation dosage to induce sterility in male melon flies are not available. Therefore, the current study aimed to evaluate the effects of different irradiation doses on critical performance aspects of male *B. cucurbitae*, specifically survival rates, flight ability, reproductive capacity, and sexual competitiveness, in comparison with their wild counterparts from local strains of melon flies. This evaluation is essential as the foundation for the establishment of a successful SIT-based pest control approach to eradicate melon flies from agricultural fields in Sri Lanka.

## 2. Materials and Methods

### 2.1. Establishment of a B. cucurbitae Colony

Adult melon fly (*B. cucurbitae*) surveillance was conducted in the Mahiyanganaya area, located within the district of Kandy, Sri Lanka (latitude: 7.484665 N, longitude: 80.927899 E) from April to May 2021. Infected gherkin (*Cu. anguria*) fruits were collected from the field and transported to the indoor insectary of HJS Condiments Limited, Alauwa, for colonization and mass rearing of *B. cucurbitae*. In the laboratory, infected gherkins were placed in colony-maintaining cages (45 cm in height × 45 cm in width × 45 cm in length) and monitored until adult emergence. Emerged adult melon flies were isolated and identified morphologically up to the species level by well-trained entomologists. Eggs laid by a single, separated female *B. cucurbitae* fly were used to establish a colony. Adults were maintained in rearing cages with mesh screening on top, under standard laboratory conditions: a 12:12 (light to dark) photoperiod, 27 ± 2 °C, and 75 ± 5% relative humidity. Emerged adult flies were fed with a fresh 20% sugar solution (prepared by mixing 20 g of sucrose and 2 mL of Vitamin B in 98 mL of distilled water). To induce oviposition, a solution of hydrolyzed yeast (1 g), sucrose (3 g), and deionized water (10 mL) was placed in a thin sponge with a tray [9].

Eggs deposited by *B. cucurbitae* were collected from the adult-rearing cage, where a petri dish containing water was positioned. The eggs were gathered daily and transferred into a plastic tray containing an artificial diet. The newly hatched larvae were fed with a mill feed diet composed of a homogenized mixture of 28% bran, 7% yeast, 13% sugar, 0.28% sodium benzoate, 1.7% hydrochloric acid, and 50.2% water [10]. Larvae were raised in small plastic trays (12.5 × 7 × 3 cm in dimensions) filled with 120 g of the prepared diet. Each tray was covered with a glass piece wrapped in a dark plastic sheet for 24 h to maintain a relative humidity of 80 to 90%. These trays were then placed inside larger plastic trays (20 × 20 × 10 cm in dimensions) lined with a thin layer of clean sand and covered with muslin cloth. After 7 days, the sand was sieved to collect the developed pupae. The collected pupae were segregated into males and females based on their body lengths (females: 8.05–8.74 mm; males: 9.50–10.2 mm), as described by Pradhan et al. [11]. Pupae were then placed inside separate adult-rearing cages until adult emergence.

### 2.2. Gamma Radiation Treatment

Two days before emergence (after 24 h), male pupae were separated and transported to the Horticultural Crop Research and Development Institute, Gannoruwa, Sri Lanka. The selected male pupae of *B. cucurbitae* were irradiated at eight different radiation doses, namely 0 (control), 50, 60, 70, 80, 90, 100, and 110 Gy, using an irradiator (Gammacell 220, Atomic Energy of Canada Ltd., Ottawa, Canada; Co-60) under dry conditions. The irradiation dosages were determined using the Fricke dosimetry system [12]. A total of 100 *B. cucurbitae* pupae were irradiated under each dosage, and the entire experiment was triplicated.

### 2.3. Effects of Irradiation on Pupal Mortality

After irradiation, 70 pupae exposed to different irradiation doses were transferred into separate 250 mL plastic cups and were placed inside separate adult-rearing cages, allowing adult *B. cucurbitae* to emerge under a 12:12 (light to dark) cycle under standard conditions (at 27 ± 2 °C and 75 ± 5% humidity). After 72 h, emerged adults were counted, and the total surviving adults of *B. cucurbitae* in each irradiation dose were reported.

### 2.4. Flight Ability

For each dose treatment and the control, 30 pupae were placed separately in 90 mm petri dishes. Over each dish, a black tube measuring 100 mm in height and 94 mm in inner diameter was positioned. The interior of the tubes was lightly coated with unscented talcum powder. A 15 mm wide section of talcum powder at the base of each tube was removed to create an additional resting area for the newly emerged melon flies. Subsequently, each set-up was placed within adult-rearing cages (45 cm in height × 45 cm in width × 45 cm in depth). The number of newly emerged adult melon flies that came out of the tubes was enumerated and recorded separately, adhering to the guidelines specified under the FAO/IAEA/USDA (2003) manual.

### 2.5. Reproductive Sterility

Within 2 days after emergence, eight male *B. cucurbitae* from each irradiation treatment were placed into separate adult-rearing cages (45 cm × 45 cm × 45 cm) with eight females, and adult flies were fed as described in Section 2.1. After a period of 15 days, gherkins were introduced into each cage, allowing *B. cucurbitae* to lay eggs over a three-day period. After the completion of the period of exposition, gherkins were collected from each cage and kept in 2 L plastic containers (dedicated for each irradiation treatment) on a vermiculite layer to facilitate larval development. The plastic containers were covered with voil. Subsequently, the total number of larvae and pupae emerging from gherkin after 15 days were enumerated to estimate the sterility [12]. The entire experiment was triplicated.

### 2.6. Effects of Irradiation on B. cucurbitae Adult Survival

Groups of 25 male *B. cucurbitae* exposed to each irradiation treatment were placed separately in adult mosquito cages (45 cm high × 45 cm wide × 45 cm deep). The cages were maintained under standard rearing conditions, while continuously providing food and water. Daily mortality of *B. cucurbitae* was recorded until the death of the last individual [12].

### 2.7. Mating Competitiveness

The degree of sterility induced by sterile males in the wild population could be assessed as the reduction in egg hatching and is referred to as the mating competitiveness index (CI) [12]. Male *B. cucurbitae* exposed to the best irradiation dose (70 Gy) was used for the following experiments conducted under laboratory and semi-field settings to evaluate the relative mating competitiveness of sterilized male *B. cucurbitae*.

### 2.8. Under Laboratory Conditions

Sterilized males (4–5 days old, irradiated at 70 Gy) and untreated males of the same age were placed in field cages measuring 45 cm × 45 cm × 45 cm, along with virgin females of identical age, at ratios of 1:1:1, 3:1:1, and 5:1:1 (sterile males to untreated males to virgin females). The total melon fly density was maintained at 30 melon flies per cage. Control experiments were conducted by introducing a 1:1 ratio of sterile males to virgin females and a separate 1:1 ratio of untreated males to virgin females into individual cages [13]. The cages were provided with food as described in Section 2.1. At 15 days of age, similarly sized gherkins were placed in each cage, allowing the flies to oviposit over a 3-day period. Following oviposition, the gherkins were transferred into 2 L plastic containers lined with vermiculite to facilitate larval development. The containers were covered with voile fabric. After 15 days, the number of larvae and pupae emerging from each treatment was counted to assess sterility. The entire experiment was triplicated.

### 2.9. Semi-Field Conditions

Three experimental ratios, namely 1:1:1, 3:1:1, and 5:1:1 (sterile males to fertile males to virgin females, n = 50), were evaluated in semi-field cages measuring 1.82 m × 1.21 m × 1.21 m, using young *B. cucurbitae* 4 to 5 days post-emergence. Control groups consisted of a 1:1 ratio of sterile males to virgin females and a 1:1 ratio of untreated males to virgin females. After a 3-day mating period, the females were transferred into adult-rearing cages (45 cm × 45 cm × 45 cm) and brought to the laboratory for further observation. The cages were provided with food as described in Section 2.1. When the adults reached 15 days of age, gherkins were placed in each cage, allowing the flies to oviposit over a 3-day period. Following oviposition, the gherkins were transferred into separate 2 L plastic containers lined with a bed of vermiculite to facilitate larval development. Each container was covered with voile fabric. After 15 days, the larvae and pupae from each treatment were counted to assess sterility. The entire experiment was conducted in triplicate.

### 2.10. Statistical Analysis

All the recorded data were entered into IBM SPSS Statistics (version 23), adhering to the standard quality control procedures. Mean values of pupal survival, flight ability, fecundity, fertility, and longevity were calculated. The mating competitiveness value (CI) was computed using the following formula [13] for *B. cucurbitae* under laboratory and semi-field conditions, separately using Equation (1). The hatch rates of the fertile and sterile control groups were used for this estimation, along with the observed egg hatch rates with a 1:1:1 ratio of sterile males to wild males to wild females.
Male competitiveness index (CI) = ((Hn-Ho)/(Ho-Hs)) × (N/S)….(1)

In this context, Hn represents the hatch rate of eggs from females mated with untreated males, while Hs corresponds to the hatch rate from females mated with sterilized males. Ho refers to the observed egg hatch rate under a 1:1:1 ratio (sterile males to wild males to wild females) of *Bactrocera cucurbitae*. Meanwhile, N is the number of untreated males, and S is the number of sterile males. The significance of the effect of radiation dose on different pupal measurements and adult performance parameters was analyzed using General Linear Modeling (GLM) followed by Tukey’s pairwise comparisons in SPSS (version 23).

## 3. Results

### 3.1. Effects of Irradiation on Pupal Survival

The highest mean pupal survival rate was reported from the *B. cucurbitae* exposed to control conditions as 96.9 ± 0.88% (Table 1). Meanwhile, the lowest pupal survival rate of 83.3 ± 2.33% was reported from the *B. cucurbitae* exposed to 110 Gy radiation. The mean pupal survival rates of *B. cucurbitae* denoted significant differences (*p* < 0.001) at a 95% level of confidence among different radiation exposure levels, as shown in Table 1. The mean pupal survival rates of *B. cucurbitae* decreased with the increasing irradiation dose. According to the post hoc analysis of the mean pupal survival rates, the adult emergence rates of *B. cucurbitae* did not denote any significant differences below the 70 Gy exposure level.

### 3.2. Effects of Irradiation on Flight Ability

The percentage flight ability scores of *B. cucurbitae* exposed to different radiation levels reported significant differences (*p* < 0.001) among the irradiation levels (Table 1). The highest percentage score for flight ability of emerged *B. cucurbitae* was observed from the control treatment as 96.7 ± 1.92%, while the *B. cucurbitae* exposed to an irradiation dose of 110 Gy reported the lowest percentage score for flight ability (64.4 ± 2.94%), as shown in Table 1. Similar to the pupal emergence rates, the flight ability scores of *B. cucurbitae* were similar up to 70 Gy.

### 3.3. Effects of Irradiation on Fecundity and Fertility

Regardless of being exposed to different irradiation levels, the mean fecundity levels of *B. cucurbitae* did not show any significant differences (*p* = 0.281), as shown in Table 1. The highest fecundity was observed in the non-irradiated *B. cucurbitae* at 19.5 ± 0.52, while the lowest fecundity was observed in *B. cucurbitae* exposed to an irradiation level of 100 Gy at 18.7 ± 0.24. Therefore, findings suggested that being exposed to irradiation did not have any impact on the fecundity of *B. cucurbitae*. In the case of fertility, significant differences were observed (*p* < 0.001) among *B. cucurbitae* exposed to different irradiation levels. The highest and lowest percentage fertility levels were reported from *B. cucurbitae* exposed to zero radiation (control) and 110 Gy, as 95.4 ± 0.48% and 0% (Table 1). The fertility rates denoted a drastically reducing trend with the increasing irradiation dose of exposure, as shown in Table 1.

### 3.4. Effects of Irradiation on Adult Survival

The number of days survived by 50% and 75% of *B. cucurbitae* population exposed to different irradiation doses are shown in Figure 1. The 50% and 75% adult longevity periods gradually decreased with the increasing irradiation dosage. The highest and lowest longevities until reaching 50% mortality of the population were reported from the *B. cucurbitae* exposed to zero irradiation and 110 Gy, as 37.3 ± 1.28 days and 15.6 ± 1.3 days, respectively (Figure 1). A similar trend was also observed for 75% adult longevity, where *B. cucurbitae* exposed to control treatment reported the highest survival period of 44 ± 2.3 days, while 110 Gy irradiation dose denoted the lowest survival period (21.7 ± 3.48 days). Both 50% and 75% longevity periods denoted significant differences (*p* < 0.001) among the exposed irradiation doses at a 5% level of significance.

### 3.5. Mating Competitiveness Index (CI) of Sterile Male Melon Flies

The percentage hatching rates of irradiated and untreated treatments were significantly different (*p* < 0.05). However, the mean hatch rates (%) of the 1:1:1 ratio did not denote any significant difference from the 3:1:1 ratio (*p* > 0.05). Meanwhile, the mean hatch rate reported by the 5:1:1 treatment was significantly different from the aforementioned treatments (*p* < 0.05). A CI value of 0.56 was reported from the sterilized males, suggesting they were approximately half as competitive as their untreated counterparts in laboratory settings (Table 2). In semi-field conditions, the hatching rates (%) significantly differed between sterile males and untreated controls (*p* < 0.05). However, percentage hatching rates were similar between the 1:1:1 and 3:1:1 ratios (*p* > 0.05). Similar to laboratory conditions, the sterilized males exhibited a CI value of 0.50, indicating that they were also approximately half as competitive as untreated males in semi-field conditions (Table 3). The fertile control with the untreated *B. cucurbitae* reported mean hatching rates of 80.1 ± 2.4% and 78.9 ± 3.1% under laboratory and semi-field conditions, separately.

## 4. Discussion

Among different vector control strategies used to manage melon fly attacks, the SIT-based pest control strategy has emerged as a novel but innovative approach, which includes releasing a large number of sterile males to suppress the wild population [14]. This study attempted to assess the effect of irradiation on the fertility and other performance parameters of male melon flies under laboratory and semi-field settings to determine a threshold dose, which induces the optimum sterility without any deleterious impacts on the performance characteristics. The findings of this study have laid the groundwork for implementing a SIT-based melon fly control programme in Sri Lanka.

The observed variations in male pupal survival rates of *B. cucurbitae* (melon fly) exposed to different radiation exposure levels are crucial for understanding the actual potential of utilizing SIT-based strategies to control melon flies in Sri Lanka. The highest pupal survival rate was recorded from the control treatment, while the 110 Gy radiation level resulted in the lowest pupal survival rate, proving the significant impact of irradiation on pupal viability. These findings corroborate the outcomes of previous studies, which have demonstrated the susceptibility of *B. cucurbitae* to radiation-induced mortality [9,12]. The decrease in pupal survival rates with increasing radiation doses in the current study perfectly aligns with the principle of radiation-induced sterility, wherein higher doses lead to greater damage to reproductive tissues, ultimately reducing the ability of pupae to successfully develop into adults [12,14].

Furthermore, the significant differences observed in male pupal survival rates among radiation exposure levels underscore the dose-dependent nature of the sterilizing effect [14]. These findings are consistent with previous research on the effects of irradiation on insect sterilization, emphasizing the importance of dose optimization to maximize sterilization efficacy, while minimizing detrimental effects on other performance characteristics of the target species [14,15,16]. According to the findings of this study, exposure of *B. cucurbitae* to 70 Gy radiation or below has not reported any significant differences in adult emergence rates, which are similar to the control. Therefore, exposure to a radiation level of 70 Gy or below has not significantly influenced the adult emergence of *B. cucurbitae*. A recent study conducted by Panduranga *et al.* [12] in India has also revealed a similar trend in *B. cucurbitae* exposed to irradiation levels of 0 to 50 Gy. However, the highest adult emergence of *B. cucurbitae* (80.7 ± 4.38%) reported by Panduranga *et al.* [12] was lower than the adult emergence observed at 70 Gy treatment (94.0 ± 1.25%) in the current study.

Understanding the impact of irradiation on the flight ability of *B. cucurbitae* is crucial in designing SIT-based control strategies, which depict the mobility, dispersal ability, and competitiveness of emerged adults. The highest flight ability was recorded under control conditions, indicating the robust flight capabilities in untreated (non-irradiated) individuals, reflecting the natural mobility of *B. cucurbitae*. In contrast, exposure to a radiation dose of 110 Gy resulted in the lowest flight ability score, indicating a substantial impairment of flight capacity in irradiated individuals. Similar to the observations of the current studies, numerous studies have emphasized the negative effects of irradiation on flight performance in various insect species, including melon flies and fruit flies [17,18,19]. The occurrence of radiation-induced damage to flight muscles, neural pathways, and sensory organs has been identified as the major reason behind this severely compromised flight ability [18,19]. Notably, the similarity in flight ability scores among *B. cucurbitae* exposed to radiation doses below 70 Gy suggests a dose threshold, below which irradiation does not significantly impair flight performance. This threshold phenomenon has been observed in other insect species subjected to irradiation-based sterilization techniques and may reflect the resilience of certain physiological mechanisms at lower radiation doses [12,20].

Despite differences in radiation exposure, the fecundity of female *B. cucurbitae* remained relatively stable, with no statistically significant variations detected among treatment groups. The consistent mean fecundity levels observed among female *B. cucurbitae* mated with irradiated males in various irradiation levels suggest that irradiation does not significantly affect the reproductive output, which is in line with the findings of some previous studies [20]. However, several previous studies have reported a sudden reduction in fecundity after a certain radiation threshold. For instance, a recent study conducted in India has reported a notable reduction in fecundity by more than 50% when females were mated with males exposed to >50 Gy radiation [12]. In contrast to fecundity, significant differences in fertility were observed among female *B. cucurbitae* mated with males irradiated with different irradiation levels, indicating a dose-dependent impact on the reproductive success of male flies. The highest fertility rates were recorded in non-irradiated individuals (control), whereas exposure to higher radiation doses resulted in substantially reduced fertility levels. This decline in fertility with the increasing irradiation dose is consistent with the sterilizing effects of radiation on reproductive tissues, which can impair the ability of insects to produce viable offsprings [12].

Exposing male *B. cucurbitae* to radiation can induce various anatomical and physiological changes that can significantly impact their reproductive capabilities and overall fitness [21]. Previous studies have demonstrated that radiation exposure leads to alterations in both external morphology and internal reproductive organs in male flies. Exposure to irradiation can cause damage to the testes of male insects, resulting in abnormalities such as vacuolization, disintegration of germ cells, and degeneration of seminal vesicles [12,22,23]. Additionally, radiation-induced damage could extend to the accessory glands, which play a crucial role in sperm viability and fertility. Further, exposure to irradiation could reduce the size and functionality of accessory glands, leading to decreased sperm production and impaired mating performance [22]. In addition, radiation exposure can disrupt hormonal regulation pathways involved in reproductive processes, including spermatogenesis and mating behaviour [23,24,25].

Understanding the relationship between irradiation dose and adult longevity is crucial for optimizing the implementation of SIT-based approaches. The adult longevity of *B. cucurbitae* in response to varying irradiation doses provided valuable insights into the impact of radiation exposure on their lifespan. The gradual decrement in both 50% and 75% longevity periods with increasing irradiation dosage underscores the dose-dependent nature of the detrimental effects of radiation on adult survival. The highest longevity periods, until reaching 50% and 75% mortality, were recorded from non-irradiated individuals (control), which highlights the natural lifespan of *B. cucurbitae* under optimal conditions. In contrast, *B. cucurbitae* exposed to 110 Gy exhibited the lowest survival durations. The significant differences observed in both 50% and 75% longevity periods among irradiation doses further emphasized the impact of radiation exposure on adult survival. These results were consistent with similar studies conducted on other fruit fly species [12,16].

The significant differences observed in hatch rates between irradiated and untreated controls confirm the efficacy of the SIT approach in reducing the reproductive potential of the melon fly population. The absence of a significant difference in hatch rates between the 1:1:1 and 3:1:1 ratios (sterile males to untreated males to virgin females) under both laboratory and semi-field conditions suggests that the release of sterile *B. cucurbitae* males, even at lower ratios, would be sufficient for notable population suppression, thereby potentially optimizing the cost-effectiveness of SIT applications. However, the lower competitiveness index (CI) of sterile males under both laboratory (0.56) and semi-field (0.50) conditions warranted the need for continuous release of *B. cucurbitae* and optimization of sterilization techniques to enhance the overall efficacy of SIT programmes in field settings. Usually, the CI reflects the ability of sterile males to compete with the wild males of the target population for mating, thereby providing an assessment of the degree of sterility produced by sterile males in the wild population [12].

The findings of the current study revealed that the overall competitiveness of sterile *B. cucurbitae* males was lower than that of wild males. At the physiological level, exposure to radiation triggers oxidative stress responses and disrupts cellular homeostasis in male melon flies. This radiation-induced oxidative stress could lead to the accumulation of Reactive Oxygen Species (ROS) within cells, leading to oxidative damage to lipids, proteins, and DNA [22,23]. This oxidative damage can further impair cellular functions and contribute to the overall decline in male fitness observed following exposure to irradiation. The same observation has also been reported in previous studies for a variety of insects, such as *Anastrepha ludens* [26], *Bactrocera tryoni* [27,28], and *C. capitate* [29]. On the contrary, exposing the male *B. cucurbitae* to lower irradiation doses may avoid such a reduction in fitness and increase the competitiveness against wild males. However, such situations may enhance the risk of residual fertility of *B. cucurbitae*, thereby possibly exacerbating melon fly outbreaks. Therefore, a trade-off should be maintained between the reduction in residual fertility of irradiated melon flies and producing highly competitive sterile males.

The observed trends in decreasing fertility rates with increasing irradiation dose are evidence for the potential of the SIT to suppress *B. cucurbitae* populations in Sri Lanka, offering a sustainable and environmentally friendly approach to pest management [1]. Assessment of the trade-offs between sterilization efficacy, adult survival, and fitness is essential to develop targeted irradiation protocols that maximize the *B. cucurbitae* suppression, while minimizing unintended consequences on non-target [1,12]. These findings provide valuable insights into the practical implementation of the SIT for melon fly control in Sri Lanka, evidencing its potential as a sustainable and environmentally friendly pest management strategy in Sri Lanka and beyond.

The SIT offers several advantages as a pest management strategy, including its environmentally friendly nature, species-specific targeting, and ability to reduce reliance on chemical pesticides, thereby mitigating their associated ecological and health risks [30]. The SIT is particularly effective in isolated environments, such as islands, where the risk of re-infestation is minimal, and in species like *B. cucurbitae*, where females generally mate only once, enhancing the method’s effectiveness [3]. However, the SIT also has limitations and prerequisites for successful application. A critical requirement is the availability of a robust mass-rearing facility to produce large quantities of sterile males without compromising their fitness and competitiveness [31]. Additionally, effective SIT implementation necessitates a thorough knowledge of the target species’ biology, including mating behaviours, dispersal patterns, and population dynamics [31]. For instance, understanding the mating competitiveness of sterile males relative to wild males is essential to ensure effective population suppression [32]. These considerations highlight the importance of integrating biological insights with SIT protocols to maximize the efficacy and sustainability of this pest management approach in diverse ecological contexts.

## 5. Conclusions

Significant variations were observed in the performance parameters of male *B. cucurbitae* after being exposed to different irradiation levels. The pupal survival, flight ability, fertility rates, and adult longevity denoted significant reductions against the increasing irradiation dose. Based on the overall performance, *B. cucurbitae* male pupae (48 h before adult emergence) exposed to a 70 Gy radiation dose reported a higher pupal survival rate (94.0 ± 1.25%) and a flight ability (93.3 ± 1.88%), with a notably lower fertility (0.2 ± 0.05%). Further, the mean longevity until a 50% reduction in the population was 38 ± 1.5 days, which was satisfactory. However, the competitiveness of the irradiated *B. cucurbitae* was lower than that of the wild males, which is a mandatory sacrifice required to avoid any potential residual fertility. In this connection, 70 Gy could be recommended as the optimum irradiation dose to produce sterile *B. cucurbitae* males for the management of melon fly outbreaks in Sri Lanka, while maintaining a satisfactory level of male fitness along with the minimum residual fertility. The current findings strongly warrant the potential of using the SIT-based strategy to suppress *B. cucurbitae* populations in Sri Lanka in a sustainable and eco-friendly manner. Further studies should be conducted to evaluate the competitiveness and fitness of irradiated *B. cucurbitae* males under field conditions to optimize the SIT strategy prior to implementation.

## Figures and Tables

**Figure 1 insects-16-00021-f001:**
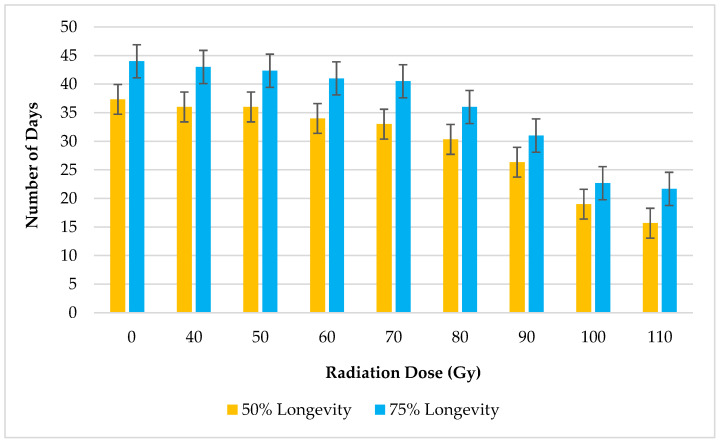
Adult longevity of *B. cucurbitae* exposed to different irradiation doses.

**Table 1 insects-16-00021-t001:** Percentage pupal survival, flight ability, fecundity, and fertility of *B. cucurbitae* exposed to different radiation levels.

Irradiation Dose (Gy)	Pupal Survival (%)	Flight Ability (%)	Fecundity	Fertility (%)
0	96.9 ± 0.88 ^a^	96.7 ± 1.92 ^a^	19.5 ± 0.52 ^a^	95.4 ± 0.48 ^a^
40	96.1 ± 0.98 ^a^	95.6 ± 1.11 ^a^	19.3 ± 0.44 ^a^	22.9 ± 0.96 ^b^
50	95.9 ± 1.88 ^a^	93.3 ± 1.92 ^a^	19.4 ± 0.19 ^a^	13.1 ± 1.71 ^b^
60	95.0 ± 1.53 ^a^	92.2 ± 4.11 ^a^	19.1 ± 0.61 ^a^	6.3 ± 0.37 ^c^
70	94.0 ± 1.25 ^a^	93.3 ± 1.88 ^a^	19.1 ± 0.37 ^a^	0.2 ± 0.05 ^d^
80	87.7 ± 0.88 ^b^	75.6 ± 2.22 ^b^	19.0 ± 0.68 ^a^	0.0
90	87.3 ± 0.33 ^b^	72.2 ± 2.45 ^b^	19.0 ± 0.18 ^a^	0.0
100	86.7 ± 1.76 ^b^	65.8 ± 1.11 ^c^	18.9 ± 0.11 ^a^	0.0
110	83.3 ± 2.33 ^b^	64.4 ± 2.94 ^c^	18.7 ± 0.24 ^a^	0.0

Note: Values are mean ± Standard Error. Different superscript letters in each column depict significant differences among mean, as suggested by the GLM, followed by Tukey’s pairwise comparison at the 95% level of confidence.

**Table 2 insects-16-00021-t002:** Competitiveness index (CI) of sterile *B. cucurbitae* males under laboratory conditions measured with different ratios of sterile to untreated males.

Ratio	Hatch Rates (%)	Competitiveness Index (CI)
Fertile Control	80.1 ± 2.4 ^a^	
Sterile Control	0.7 ± 0.1 ^b^	
1:1:1	55.4 ± 2.2 ^c^	0.56
3:1:1	47.2 ± 1.8 ^c^	
5:1:1	31.9 ± 2.5 ^d^	

Note: Values are mean ± Standard Error. Different superscript letters in each column depict significant differences among mean, as suggested by the GLM, followed by Tukey’s pairwise comparison at the 95% level of confidence.

**Table 3 insects-16-00021-t003:** Competitiveness index (CI) of sterile *B. cucurbitae* males under semi-field conditions in large cages, measured with different ratios of sterile to untreated males.

Ratio	Hatch Rates (%)	Competitiveness Index (CI)
Fertile Control	78.9 ± 3.1 ^a^	
Sterile Control	0.8 ± 0.1 ^b^	
1:1:1	50.5 ± 2.1 ^c^	0.50
3:1:1	47.7 ± 1.6 ^c^	
5:1:1	32.8 ± 3.3 ^d^	

Note: Values are mean ± Standard Error. Different superscript letters in each column depict significant differences among mean, as suggested by the GLM, followed by Tukey’s pairwise comparison at the 95% level of confidence.

## Data Availability

The raw data supporting the conclusions of this article will be made available by the authors upon request.

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
