# Peer review of "Evaluation of the Sterile Insect Technique for the Control of the Melon Fly (*Bactrocera cucurbitae*) Under Laboratory and Semi-Field Conditions in Sri Lanka"

_insects, 2024, doi:10.3390/insects16010021_

Round 1
Reviewer 1 Report
Comments and Suggestions for Authors
The study was aimed at Development of Sterile Insect Technique based approach to control the melon fly (Bactrocera cucurbitae) in Sri Lanka.
The research question is interesting and shows some promising results. The methodology is well designed, and the results are presented clearly. However, the manuscript can be accepted for publication with minor revisions.
Line 25: change ‘papa’ to “pupae”
Line 118: add “2. Material and methods” section
Line 172: ‘The cages were provided with water-soaked cotton and the solid food as described above (section …..?)’. please provide the section number.
Line 273: change ‘100 Gy’ to “110 Gy” in consistence with the results from Table 1.
Author Response
Reply to Reviewers
Manuscript No: Insects-3333142
Authors would like to thank all the reviewers for their valuable comments, which contributed immensely to improve the quality of the manuscript. We would like to emphasize that all suggestive changes have been addressed as much as possible. Changes made in the manuscript have been highlighted for your convenience.
- Reviewer 1
The study was aimed at Development of Sterile Insect Technique based approach to control the melon fly (Bactrocera cucurbitae) in Sri Lanka. The research question is interesting and shows some promising results. The methodology is well designed, and the results are presented clearly. The manuscript can be accepted for publication with minor revisions.
- Comment 1
Line 25: change ‘papa’ to “pupae”.
Response
The comment is welcome. The term "papa" was corrected to "pupae" on line 25 as suggested.
- Comment 2
Line 118: add “2. Material and Methods” section.
Response
Thank you for the suggestion. The revision was done as requested.
- Comment 3
Line 172: ‘The cages were provided with water-soaked cotton and the solid food as described above (section …..?)’. please provide the section number.
Response
The authors are grateful for pointing this matter. The sentence has been updated to specify the section number as follows.
“Within two days after emergence, 8 male B. cucurbitae from each irradiation treatment were placed into separate adult rearing cages (45 cm high X 45 cm wide X 45 cm deep) with 8 females and adult flies were fed as described under section 2.1.”
- Comment 4
Line 273: change ‘100 Gy’ to “110 Gy” in consistence with the results from Table 1.
Response
Thank you for your observation. The value "100 Gy" was changed to "110 Gy" on line 273 to ensure consistency with the results presented in Table 1.
Reviewer 2 Report
Comments and Suggestions for Authors
The manuscript aimed to investigate parameters that could serve as a foundation for the future application of SIT against a significant vegetable crop pest in Sri Lanka, potentially revolutionizing pest management strategies.
It's important to ensure that the keywords should be distinct from the title, as this can significantly boost reader engagement and the manuscript's discoverability.
The title of the Material and Methods Section should be included.
In the Establishment of a B. cucurbitae Colony section. The others should include some details, such as latitude and longitude, the species and cultivar from which the insects were collected, and the date of collection. In this Section, the number of hours of light and darkness provided to the insects should be included.
Dissecting the dead pupae to assess any deformations could provide valuable insights. Authors, your contribution in this area is highly valued and could significantly advance our understanding of the subject.
The statistical analysis adopted is elementary, and the authors must use methods of comparing means.
The authors should have extracted the reproductive system under a stereomicroscope, following the dissection procedure indicated by Chou et al. (2012). The following biometric parameters should have been recorded for the ovaries: length, from the anterior end of the germarium to the calyx area, and width, taken from the anterior end of the vitellarium. In addition, they should have calculated the ovarian index by multiplying the ovarian length by the ovarian width, as Chou et al. (2012) suggested. Likewise, the testicular biometric parameters should have been recorded: length, from the apical region to the vas deferens, and width, taken from the spermatid region. We also calculated the testicular index by multiplying the testicular length by the testicular width.
- CHOU M-Y, MAU RFL, JANG EB, VARGAS RI & PINERO JC. 2012. Morphological features of the ovaries during oogenesis of the Oriental fruit fly, Bactrocera dorsalis, in relation to the physiological state. J Insect Sci 12: 1-12.

Author Response
Reply to Reviewers
Manuscript No: Insects-3333142
Authors would like to thank all the reviewers for their valuable comments, which contributed immensely to improve the quality of the manuscript. We would like to emphasize that all suggestive changes have been addressed as much as possible. Changes made in the manuscript have been highlighted for your convenience.
- Reviewer 2
The manuscript aimed to investigate parameters that could serve as a foundation for the future application of SIT against a significant vegetable crop pest in Sri Lanka, potentially revolutionizing pest management strategies.
- Comment 1
It's important to ensure that the keywords should be distinct from the title, as this can significantly boost reader engagement and the manuscript's discoverability.
Response
The authors welcome the comment. We had to revise the title based on the comments of the Reviewer 3. Therefore, following the revision of the title, we have reviewed and updated the keywords to ensure they are distinct from the title. The updated keywords now complement the title, while providing additional terms that enhance the manuscript's discoverability and relevance for readers and researchers. We appreciate your constructive feedback.
- Comment 2
The title of the “Material and Methods” Section should be included.
Response
Thank you for the suggestion. The revision was done as requested.
- Comment 3
In the Establishment of a B. cucurbitae Colony section. The authors should include some details, such as latitude and longitude, the species and cultivar from which the insects were collected, and the date of collection. In this Section, the number of hours of light and darkness provided to the insects should be included.
Response
Thank you for your valuable feedback. Based on your suggestions, the methodology section has been revised as follows.
“Adult melon fly (B. cucurbitae) surveillance was conducted in the Mahiyanganaya area, located within the district of Kandy, Sri Lanka (latitude: 7.484665 N, longitude: 80.927899 E) during April to May 2021. Infected gherkin (Cu. anguria) fruits were collected from the field and transported to the indoor insectary of HJS Condiments Limited, Alauwa, for colonization and mass-rearing of B. cucurbitae. In the laboratory, infected gherkins were placed in colony-maintaining cages (45 cm in height × 45 cm in width × 45 cm in length) and monitored until adult emergence. Emerged adult melon flies were isolated and identified morphologically up to the species level by well-trained entomologists. Eggs laid by a single, separated female B. cucurbitae fly were used to establish a colony. Adults were maintained in rearing cages with mesh screening on top, under standard laboratory conditions: a 12:12 (light: dark) photoperiod, 27 ± 2 °C, and 75 ± 5% relative humidity. Emerged adult flies were fed a fresh 20% sugar solution (prepared by mixing 20 g of sucrose and 2 mL of Vitamin B in 98 mL of distilled water). To induce oviposition, a solution of hydrolyzed yeast (1 g), sucrose (3 g), and deionized water (10 mL) was placed in a thin sponge with a tray [9].”
- Comment 4
Dissecting the dead pupae to assess any deformations could provide valuable insights. Authors, your contribution in this area is highly valued and could significantly advance our understanding of the subject.
Response
We sincerely appreciate your valuable suggestion regarding the dissection of dead pupae to assess potential deformations. We agree that such approach could provide critical insights into the understanding of the effects of irradiation on the developmental stages of B. cucurbitae. However, in the current study, we did not include this physiological analysis due to limitations in resources and time. Our primary focus was to evaluate the key performance parameters of B. cucurbitae such as survival, flight ability, fecundity, fertility, adult longevity, and mating competitiveness, which are critical for establishing an effective Sterile Insect Technique (SIT) protocol in Sri Lanka. However, we acknowledge the significance of your suggestion and plan to incorporate the dissection and detailed morphological analysis of dead pupae in our future publications. This will allow us to further explore and validate the underlying mechanisms that may affect the pupal development post-irradiation.
We believe the findings presented in this manuscript still provide a robust foundation for implementation of SIT based melon fly controlling programme in Sri Lanka and significantly contribute to the current knowledge base. Further, please note that many similar studies have also worked with this scope. We hope the scope of this study is deemed adequate for publication in its present form, with the potential for extended analysis in subsequent publications.
- Comment 5
The statistical analysis adopted is elementary, and the authors must use methods of comparing means.
Response
The authors welcome the comment. However, as clearly stated under the section 2.10, the General Linear Modeling (GLM) followed by Tukey’s pair-wise comparisons (in the form of ANOVA) was used for comparing the means of pupal measurements and adult performance parameters. To the best of authors knowledge, GLM is not an elementary analysis and is used as a standard statistical analysis technique for mean comparison under these situations. Many similar articles have also employed the same approach. The relevant section in the manuscript is as follows.
“The significance of the effect of radiation dose on different pupal measurements and adult performance parameters was analyzed using the General Linear Modeling (GLM) followed by Tukey’s pair-wise comparisons in SPSS (version 23).”
Further, please note that the other two reviewers have not raised any concern regarding the use of GLM or statistical analysis. However, the authors are willing to consider any other alternative statistical techniques suggested by the respected reviewer.
- Comment 6
The authors should have extracted the reproductive system under a stereomicroscope, following the dissection procedure indicated by Chou et al. (2012). The following biometric parameters should have been recorded for the ovaries: length, from the anterior end of the germarium to the calyx area, and width, taken from the anterior end of the vitellarium. In addition, they should have calculated the ovarian index by multiplying the ovarian length by the ovarian width, as Chou et al. (2012) suggested. Likewise, the testicular biometric parameters should have been recorded: length, from the apical region to the vas deferens, and width, taken from the spermatid region. We also calculated the testicular index by multiplying the testicular length by the testicular width.
Chou MY, Mau RFL, Jang Eb, Vargas RI & Pinero Jc. 2012. Morphological features of the ovaries during oogenesis of the Oriental fruit fly, Bactrocera dorsalis, in relation to the physiological state. J Insect Sci 12: 1-12.
Response
We appreciate your insightful suggestion regarding the use of the dissection procedure by Chou et al. (2012) to extract and measure the reproductive system, as well as the calculation of ovarian and testicular indices. We acknowledge that incorporating such biometric data would provide additional valuable insights into the physiological effects of irradiation on B. cucurbitae.
However, the primary focus of this study was to determine the best irradiation dose to establish a SIT based vector controlling programme for B. cucurbitae in Sri Lanka. Therefore, similar to many previous studies conducted in other countries, several selected parameters such as survival, flight ability, fecundity, fertility, adult longevity, and mating competitiveness were monitored by the authors. Due to resource constraints and time limitations, we were unable to include the detailed reproductive measurements described in the cited methodology. Further, please note that studying the underlying physiological mechanisms was also not the main objective of this study.
We believe that the current study provides substantial and novel contributions to the implementation of SIT for B. cucurbitae, and we respectfully request that the manuscript be considered for publication in its present form, with the potential for these additional analyses in future publications. Further, please note that both of other reviewers have not raised any concern regarding this aspect.
Reviewer 3 Report
Comments and Suggestions for Authors
1. A similarity index of 31% reported by iThenticate is considered relatively high for academic manuscripts. Ideally, the similarity rate should not exceed 20% to ensure originality and reduce concerns over potential plagiarism.
2. The current title may imply that SIT is being developed as a new method or broadly applied, which could be misleading since SIT is already established. Additionally, as your study focuses specifically on laboratory and semifield conditions, the title could be refined to reflect this scope more accurately.
A more precise title, such as "Evaluation of the Sterile Insect Technique for the Control of the Melon Fly (Bactrocera cucurbitae) under Laboratory and Semifield Conditions in Sri Lanka," may better align with the content and objectives of your study.
3. Consider adding a paragraph discussing both the advantages and disadvantages of the Sterile Insect Technique (SIT) and the specific prerequisites for its successful application. For instance, SIT is more effective in isolated environments such as islands, where re-infestation is less likely, and in target species where females preferably mate only once. In this context, it would also be beneficial to elaborate on the biology of the insect used in the study, highlighting traits relevant to the effectiveness of SIT. This would provide readers with a clearer understanding of the applicability of SIT in your research
4. Please ensure that all species names are italicized the first time they are mentioned in the manuscript and references, with the author(s) and year of description included in parentheses immediately after the species name (without italics). Bactrocera cucurbitae, Cochliomyia hominivorax, Ceratitis capitata etc.
5. Line 25: "Papa" should be revised as "pupae."
6. Line 73: "Cucumis" can be abbreviated as "Cuc." from the second mention onward in the text for consistency and brevity
7. Line 76: There appears to be a formatting issue with the font size or style in "fruits, flowers and stems." Please check and ensure consistency with the rest of the text.
8. Line 93: The abbreviation B. cucurbitae should be italicized to follow proper scientific formatting conventions
9. Lines 48, 114, 435, 472 etc: The term "Sterile Insect Technique" should be written out in full the first time it is mentioned, followed by the abbreviation "(SIT)". Subsequent mentions can use the abbreviation "SIT" consistently.
10. Line 444: [12; 16]. should be revised as [12, 16].
11. Line 465: Species names should be written in full the first time they are mentioned. A. ludens, B. tryoni etc.
12. Lines 73 and 465: The abbreviation 'C.' is used for both Cucurbita and Ceratitis, as well as for competitiveness index (C) in the text. This overlapping use may cause confusion and should be resolved. For competitiveness index, consider using CI instead of C. For genus names, ensure unique and consistent abbreviations are applied to avoid ambiguity. Please revise accordingly.
13. Please ensure that all references are formatted according to the journal’s style guidelines. There are several errors in the current reference list, including incorrect journal abbreviations, full names of journal and other formatting issues.
Comments on the Quality of English LanguageGood
Author Response
Reply to Reviewers
Manuscript No: Insects-3333142
Authors would like to thank all the reviewers for their valuable comments, which contributed immensely to improve the quality of the manuscript. We would like to emphasize that all suggestive changes have been addressed as much as possible. Changes made in the manuscript have been highlighted for your convenience.
- Reviewer 03
- Comment 1
A similarity index of 31% reported by iThenticate is considered relatively high for academic manuscripts. Ideally, the similarity rate should not exceed 20% to ensure originality and reduce concerns over potential plagiarism.
Response
Thank you for your comment regarding the similarity index. We have evaluated the plagiarism level with the Turnitin platform. According to the report, the similarity level was reduced to 20%, excluding the bibliography (references). To address this, we have thoroughly reviewed and revised the manuscript to ensure originality, removing any sections of concern. Kindly note that this 20% level includes, certain scientific names, technical terms, ethics review statements, data availability statements, affiliations etc., use of which is unavoidable. If needed authors can provide the report generated from the Turnitin platform. We appreciate your attention to this matter, which has helped us improve the quality and integrity of the manuscript.
- Comment 2
The current title may imply that SIT is being developed as a new method or broadly applied, which could be misleading since SIT is already established. Additionally, as your study focuses specifically on laboratory and semi field conditions, the title could be refined to reflect this scope more accurately. A more precise title, such as "Evaluation of the Sterile Insect Technique for the Control of the Melon Fly (Bactrocera cucurbitae) under Laboratory and Semi field Conditions in Sri Lanka," may better align with the content and objectives of your study.
Response
We appreciate your constructive feedback regarding the title of our manuscript. Following your suggestion, we have revised the title. The new title will be "Evaluation of the Sterile Insect Technique for the Control of the Melon Fly (B. cucurbitae) under Laboratory and Semi field Conditions in Sri Lanka”.
- Comment 3
Consider adding a paragraph discussing both the advantages and disadvantages of the Sterile Insect Technique (SIT) and the specific prerequisites for its successful application. For instance, SIT is more effective in isolated environments such as islands, where re-infestation is less likely, and in target species where females preferably mate only once. In this context, it would also be beneficial to elaborate on the biology of the insect used in the study, highlighting traits relevant to the effectiveness of SIT. This would provide readers with a clearer understanding of the applicability of SIT in your research.
Response
Thank you for your insightful comment regarding the need to elaborate on the advantages, limitations, and prerequisites of the Sterile Insect Technique (SIT). We have now incorporated the following paragraph in the discussion to address this comment.
“The SIT offers several advantages as a pest management strategy, including its environmentally friendly nature, species-specific targeting, and ability to reduce reliance on chemical pesticides, thereby mitigating their associated ecological and health risks [30]. SIT is particularly effective in isolated environments, such as islands, where the risk of re-infestation is minimal, and in species like B. cucurbitae, where females generally mate only once, enhancing the method’s effectiveness [31]. However, SIT also has limitations and prerequisites for successful application. A critical requirement is the availability of a robust mass-rearing facility to produce large quantities of sterile males without compromising their fitness and competitiveness [32]. Additionally, effective SIT implementation necessitates thorough knowledge of the target species' biology, including mating behaviors, dispersal patterns, and population dynamics [31]. For instance, understanding the mating competitiveness of sterile males relative to wild males is essential to ensure effective population suppression [33]. These considerations highlight the importance of integrating biological insights with SIT protocols to maximize the efficacy and sustainability of this pest management approach in diverse ecological contexts.”
- Comment 4
Please ensure that all species names are italicized the first time they are mentioned in the manuscript and references, with the author(s) and year of description included in parentheses immediately after the species name (without italics). Bactrocera cucurbitae, Cochliomyia hominivorax, Ceratitis capitata etc.
Response
We welcome the comment. The suggested revision was incorporated for all species names, where the first descriptor information was available.
- Comment 5
Line 25: "Papa" should be revised as "pupae."
Response
Thank you for pointing out this typographical error. We have revised "Papa" to "pupae" in line 25 as suggested. We appreciate your attention to detail in helping us improve the accuracy of our manuscript.
- Comment 6
Line 73: "Cucumis" can be abbreviated as "Cuc." from the second mention onward in the text for consistency and brevity.
Response
Thank you for the valuable suggestion. Based on your comments 6 and 12, the genus Cucumis was represented as ‘Cu.’, while the genus Ceratitis was represented as ‘C.’. Further, genus Cucurbita was represented as ‘Cuc.’ to avoid confusions.
- Comment 7
Line 76: There appears to be a formatting issue with the font size or style in "fruits, flowers and stems." Please check and ensure consistency with the rest of the text.
Response
Thank you for pointing out the formatting issue with "fruits, flowers and stems" on Line 76. We have reviewed and corrected the font size and style to ensure consistency with the rest of the text. We appreciate your careful attention to detail.
- Comment 8
Line 93: The abbreviation B. cucurbitae should be italicized to follow proper scientific formatting conventions.
Response
Thank you for noting the formatting issue with the abbreviation B. cucurbitae on Line 93. We have corrected it by italicizing the abbreviation to adhere to proper scientific formatting conventions.
- Comment 9
Lines 48, 114, 435, 472 etc: The term "Sterile Insect Technique" should be written out in full the first time it is mentioned, followed by the abbreviation "(SIT)". Subsequent mentions can use the abbreviation "SIT" consistently.
Response
Thank you for pointing this out. We have ensured that the term "Sterile Insect Technique" is written out in full the first time it is mentioned, followed by the abbreviation "(SIT)." It is initially mentioned in the abstract (Line no 20). In subsequent mentions throughout the manuscript, we have used the abbreviation "SIT" consistently. We appreciate your attention to detail.
- Comment 10
Line 444: [12; 16]. should be revised as [12, 16].
Response
Revised as suggested.
- Comment 11
Line 465: Species names should be written in full the first time they are mentioned. A. ludens, B. tryoni etc.
Response
The authors welcome the comment. All such instances were revised as suggested.
- Comment 12
Lines 73 and 465: The abbreviation 'C.' is used for both Cucurbita and Ceratitis, as well as for competitiveness index (C) in the text. This overlapping use may cause confusion and should be resolved. For competitiveness index, consider using CI instead of C. For genus names, ensure unique and consistent abbreviations are applied to avoid ambiguity. Please revise accordingly.
Response
We are thankful for the valuable suggestion. As suggested ‘CI’ was used to abbreviate the Competitiveness Index. The genus Cucumis was represented as ‘Cu.’, while the genus Ceratitis was represented as ‘C.’. Further, genus Cucurbita was represented as ‘Cuc.’ to avoid confusions.
- Comment 13
Please ensure that all references are formatted according to the journal’s style guidelines. There are several errors in the current reference list, including incorrect journal abbreviations, full names of journal and other formatting issues.
Response
The authors welcome the comment. All such instances were carefully revised by the authors, except for journal titles without abbreviated forms. Please note that authors are willing to address any further revisions in reference formatting.
Round 2
Reviewer 2 Report
Comments and Suggestions for Authors
The authors made most of the corrections. Some methodological suggestions were not implemented but were adequately justified by the authors. The authors should take these suggestions into account in future research, which will make the research more robust.
Reviewer 3 Report
Comments and Suggestions for Authors
The authors have adequately addressed my comments and made the necessary revisions in the manuscript
Comments on the Quality of English LanguageNo